# Interrogative Suggestibility and Ability to Give Resistant Responses in Children with Mild Intellectual Disabilities and Borderline Intellectual Functioning

**Valeria Giostra** [1] and **Monia Vagni** [1,2,*]

1   Department of Humanities, University of Urbino, 61029 Urbino, Italy; valeria.giostra@uniurb.it
2   Department of Philosophy, Social Sciences, Humanities and Education, University of Perugia, 06123 Perugia, Italy
*   Correspondence: monia.vagni@uniurb.it; Tel.: +39-072-230-5814

**Abstract:** Children with intellectual disabilities can be victims of crimes but are generally deemed less reliable in the forensic context than children without disabilities. Their deficits may cause inaccurate recall, greater memory errors, and greater suggestive vulnerability. The aim of the present study is to verify the effects of intellectual abilities on recall tasks, levels of suggestibility, vulnerability to negative social pressure, and Resistant Behavioural Responses (RBR). The study involved 120 children aged 7–16 years who were administered the GSS2 (1997) and Raven Matrices. Forty children had a diagnosis of mild intellectual disability (MID), 40 had borderline intellectual functioning (BIF), and 40 were typically developing peers. Children with MID and BIF showed more errors in distortions, inventions, and confabulations at the recall task and higher levels of suggestibility. Low IQs reduced the ability of source monitoring and led to less resistant responses to misleading questions. IQ affected resistant responses both at the first and second suggestive interview, reducing both source monitoring capabilities and the ability to manage social pressure. Age may impact the ability to provide resistant responses but only in the first suggestive interview (Yield 1), which depends more on the maturity of cognitive processes involved in interrogative suggestibility.

**Keywords:** leading questions; borderline intellectual functioning; intellectual impairment; interrogative suggestibility; confabulation; children

## 1. Introduction

Children with intellectual disabilities (ID) are particularly at risk of being victims of crime. More specifically, the literature confirms how they are more vulnerable to being maltreated (Horner-Johnson and Drum 2006; Wissink et al. 2015) or sexually abused (e.g., Sullivan and Knutson 2000; Vig and Kaminer 2002; Bryen et al. 2003; Wissink et al. 2015; Brendli et al. 2022). Therefore, it is likely that they will have to be heard in court as witnesses, which causes their reliability to become a matter of significant interest.

When a minor has to be heard as a witness, it is always important to evaluate the cognitive and social factors implied in the witnessing. This is even more crucial for children with ID. Indeed, ID impacts the cognitive functions of the individual implied in the witnessing, and, as a result, the reliability of these children's testimony is often influenced by negative stereotypes about their abilities, which may be underestimated (Ellis et al. 2017).

In particular, the literature underlines that, in the forensic context, it is always very important to assess the level of suggestibility of minors, as they are more vulnerable to leading questions than adults. Many studies examined the relationship between intellectual disability and levels of suggestibility in minors, showing results that are not always unanimous (Klemfuss and Olaguez 2020). The main purpose of this study is to clarify whether different degrees of intellectual disability may lead children to inaccurate recall and greater

suggestive vulnerability. Furthermore, it will be investigated whether ID influences the type of resistant response to leading questions according to the RBR model (Gudjonsson et al. 2021).

*1.1. Interrogative Suggestibility and Resistant Behavioural Responses Models*

Interrogative suggestibility (IS) is a psychosocial model that refers to a person's tendency to accept leading questions and negative pressure when under interrogation (Gudjonsson and Clark 1986).

The IS model pays attention both to the way in which the questions are formulated, which contain suggestive or misleading information and to the social pressure given by negative feedback. Leading questions tend to lead the interviewee to accept the suggestions, while negative feedback tends to modify the answers. Two factors compose the IS: the Yield or acceptance of leading questions and the Shift in answers after negative feedback (Gudjonsson 2003).

Gudjonsson created a tool for evaluating these factors, the Gudjonsson Suggestibility Scale (Gudjonsson 1997), which involves, after reading a story, the administration of a suggestive interview followed by negative feedback. The aim of this tool is to evaluate Yield and Shift scores. After the negative feedback, the interview is repeated in order to obtain the value of Yield 2, i.e., the number of suggestive questions accepted after the negative feedback.

According to the IS, three factors are necessary for the occurrence of a suggested response: uncertainty, interpersonal trust, and expectations of success. Uncertainty occurs because of leading questions that introduce misleading information and lead the interviewee to be uncertain about the correct answer, especially if this information appears pertinent, plausible, and compatible with one's knowledge and expectations. Interpersonal trust is the result of the interviewee or witness's belief that the interviewer's intentions are true; the expectations of success refer to the fact that most people think they are supposed to provide a clear and certain response because they are expected to know that given response.

The literature showed how interrogative suggestibility is influenced by various cognitive and socio-emotional factors. Among the cognitive factors, age is an important variable: younger children generally show higher levels of interrogative suggestibility (Ceci et al. 2007; Goodman et al. 2014), and this may be due to the incomplete development process of cognitive functioning related to interrogative suggestibility, i.e., memory, attention, executive functioning. After the age of twelve, however, the levels of suggestibility are similar to those of adults (Gudjonsson 2003). As will be better explained later, intelligence and cognitive abilities also influence interrogative suggestibility (Gudjonsson 2018).

With regard to socio-emotional factors, low levels of self-esteem and exposure to stressful and traumatic events seem to correlate with higher levels of suggestibility (Drake 2014; Drake et al. 2008; Gudjonsson et al. 2020; Vagni et al. 2021). However, these factors will not be taken into consideration in this study, which focuses mainly on the effects of intellectual disability.

Although the literature on the factors that influence suggestibility is extensive, to date, few studies (Gudjonsson et al. 2021, 2022; Vagni et al. 2023) investigated how children respond to suggestive questions and, in particular, what kind of resistant responses they can provide. In fact, faced with a leading question, a minor can answer by accepting, rejecting, or admitting "I don't know the answer". The different responses that refuse a leading question represent the Resistant Behavioural Responses (RBR; Gudjonsson et al. 2021, 2022).

When asked leading questions, an interviewee can yield to the suggestion in three main ways: (a) they can reject the suggestion by simply saying "no" (NO answers); (b) they can admit their uncertainty, saying they do not know (DK answers); (c) they can provide a direct explanation by saying, for example, that what is suggested was not mentioned or did not happen (DE answers). These three main types of resistant responses to leading

questions on Yield 1 and Yield 2 (DK, DE, and NO answers) can be readily measured by the GSS scales (Gudjonsson 1997, 2003).

Resistant Behavioral Responses (RBRs) refer to the construct of source monitoring (SMF; Johnson 1997; Johnson et al. 1993; Johnson and Raye 1981), which involves the ability to recognize where information comes from. In the specific context of interrogative suggestibility, source monitoring allows us to recognize whether the requested information corresponds to what was previously stored. People with adequate levels of source monitoring are able to reject leading questions with "No" answers. High levels of source monitoring, according to the RBR model, allow us to reject misleading information through clarifications or Direct Explanations (DE responses) (such as "This information was not there" or "It did not happen").

Starting from the RBR model, a study by Gudjonsson et al. (2022) showed that NO, DE, and DK answers are different and independent response styles that have different effects on resistance to leading questions because they are driven by different cognitive processes. DK answers may be appropriate and helpful but nevertheless could indicate a problem with "source monitoring" because of a failure to identify the discrepancy between what the interviewees observed and what was subsequently suggested to them by the interviewer (Gudjonsson et al. 2021). NO and DE answers imply correct source monitoring. However, DE answers require that the children recognize the discrepancy between what was observed and what was suggested and can, therefore, articulate an appropriate explanatory resistant response, which is possible when effective strategic source monitoring and control processes occur (Koriat et al. 2001). For these reasons, "Direct Explanation" answers tend to remain constant in repeated suggestive interviews (Vagni et al. 2023) and show a protective effect against suggestive questions and negative feedback (Gudjonsson et al. 2021). Resistant answers seem to depend on the cognitive maturity and source monitoring skills of the interviewee and not on the type of suggestive question that is asked. The relationship between RBR and intellectual vulnerability was investigated in a sample of adults (Gudjonsson and Young 2021) but not in a sample of children.

### 1.2. Intellectual Disabilities and Suggestibility

ID involves problems with general mental abilities that affect functioning in two areas: intellectual functioning (such as learning, problem-solving, judgment) and adaptive functioning (activities of daily life such as communication and independent living). There are four levels of ID: mild (IQ 70/51), moderate (IQ 50/36), severe (IQ 35–20), and profound (IQ $\leq$ 19). Although it is not taken into consideration in the current diagnostic systems, it is noteworthy to mention also borderline intellectual functioning (BIF), which is included in the DSM 5TR and is commonly defined as scoring between 1 and 2 standard deviations below the mean on standardized tests of intelligence, typically equivalent to an IQ of less than 85.

ID impacts the various cognitive functions of the individual implied in the witnessing. The literature shows that in children with ID, there are more deficits in attentional and executive functions than in typically developing children (TD; Carrasco et al. 2005; Leyfer et al. 2006; Rhodes et al. 2011; Kreutzer et al. 2011), and this significantly affects their learning abilities and behavior (Deutsch et al. 2008; Simonoff et al. 2007). Working memory also appears to be less efficient in children and adolescents with ID (Henry 2001; Schuchardt et al. 2011).

The literature (Peltopuro et al. 2014) confirmed that even children with BIF show impairments in working memory (Alloway 2010; Schuchardt et al. 2011); limits in executive functions; deficits in processing speed (Bonifacci and Snowling 2008); poor attention and concentration skills and concrete thinking; and a lack of the ability to generalize knowledge, strategies, and learning (Ninivaggi 2005).

Despite this, the literature confirmed that children with mild ID are able to describe their experiences as truthfully as children with typical development, especially when the interview is conducted relatively soon after the event. On the contrary, children with more

severe cognitive impairment can provide valid information but fewer details than children without ID (Pipe et al. 2004; Salmon and Pipe 2000). Cognitive difficulties can lead to an autobiographical memory poor in detail and this seems to be associated with a greater vulnerability to suggestibility factors (Vagni et al. 2024).

The type of questions used in the interview is also very important: a study by Brown et al. (2015) highlighted that if open-ended questions are used, children with ID are able to be as accurate as individuals without cognitive impairments, although they tend to provide less information and fewer details (Gordon et al. 1994; Henry and Gudjonsson 1999, 2003; Agnew and Powell 2004; Henry and Gudjonsson 2007). The use of specific questions, on the other hand, negatively affects the level of accuracy.

In the forensic context, it is likely that a child will be subjected to leading questions.

Many studies examined the relationship between intellectual disability and levels of suggestibility, showing results that are not always unanimous (Klemfuss and Olaguez 2020).

Children with ID tend to be as suggestible as mental age-matched TD children (London et al. 2013) and more suggestible than chronological age-matched TD children (Bowles and Sharman 2014).

In a study by Gudjonsson (2003), lower IQs were positively correlated with high levels of interrogative suggestibility at GSS2, particularly with regard to Yield 1, which seems to be more associated with cognitive factors than Shift. Miles et al. (2007) found that ID children showed higher scores on the Gudjonsson Suggestibility Scale but not higher levels of delayed suggestibility. However, Robinson and McGuire (2006) did not find significant differences in GSS-2 scores between children with mild ID and the control group, while Gudjonsson and Young (2010) found a very weak correlation between intelligence and GSS.

Milne et al. (2013) found that ID children are more suggestible than TD children when asked misleading questions about a video. Henry and Gudjonsson (2007) found that children whose ID was identified based on verbal intelligence were more suggestible than chronological age-matched TD children. In a study by Vagni et al. (2021), ID was significantly associated with a higher production of distortions and fabrications, but only children with severe ID showed a greater level of interrogative suggestibility.

According to Klemfuss and Olaguez (2020), the links between ID and suggestibility remain inconsistent, and ID itself does not appear to predict suggestibility. This may be due to the fact that the various studies use different methodological approaches: some studies examined continuous differences in intelligence within the population, whereas other studies compared suggestibility in children with or without intellectual disability. Furthermore, the various studies use different tools for assessing intelligence and suggestibility.

The attention of studies conducted on children with intellectual vulnerabilities seems to be focused more on children with ID and not on children with BIF. Therefore, the effects of BIF on suggestibility and the ability to produce resistant responses to misleading questions remain relatively unknown.

The present study aims to make a contribution to clarifying this relationship, involving children with different degrees of intellectual functioning. Furthermore, not only suggestibility scores will be taken into consideration but also memory errors such as distortions, fabrications, and confabulations. Moreover, the attention paid to the production of memory errors at GSS2 by children with ID also seems to be limited so far. A previous study highlighted a negative correlation between IQ and memory errors (Vagni et al. 2021). However, in this study, no comparisons were made because groups of children with different levels of intellectual functioning were not involved.

Therefore, it appears important to try to clarify this relationship also with regard to the RBR model. Gudjonsson (2018) highlighted that borderline intelligence was linked to higher suggestibility levels, and this could be due to children's difficulties in executive functions and source monitoring (Lee and Kim 2022; Lee and Shin 2023). In adults with intellectual disabilities, Gudjonsson and Young (2021) found that their low capacity in the social cognitive mechanism associated with low confidence and trust in one's memory limited more resistant responses such as "Direct Explanation" and "I don't know".

*1.3. The Present Study*

In this paper, we propose to investigate the effects of IQ on children with mild intellectual disabilities, borderline intellectual functioning, and typically developing children on memory tasks and interrogative suggestibility levels.

Previous studies investigated the relationship between intelligence and suggestibility, but in this study, both mild intellectual disabilities and borderline intellectual functioning will be taken into consideration and will be compared with typical development using a metric and standardized instrument (GSS2). The main originality of the study is given by the fact that, for the first time, the RBR model was applied to the responses of children with intellectual disabilities to Gudjonsson's suggestibility scale.

In fact, no study considered memory errors, Immediate Recall, interrogative suggestibility by GSS2 and resistant responses in children with different degrees of intellectual impairment.

Based on the reviewed literature, we formulated the following set of hypotheses:

**Hypothesis 1.** *MID and BIF show more memory errors and immediate suggestibility levels than typically developing children;*

**Hypothesis 2.** *IQ levels lead to poorer Immediate Recall, a greater number of memory errors and higher interrogative suggestibility scores;*

**Hypothesis 3.** *Lower IQ levels reduce resistant responses with higher source monitoring capacity, such as "NO" and "Direct Explanation" responses.*

## 2. Materials and Methods

*2.1. Participants*

The sample included 120 participants (49.2% female; 50.8% male) aged between 7 and 16 years ($M = 11.49$ and SD = 2.44). All participants were Italian and were recruited from child neuropsychiatry services for checks on their cognitive development.

Forty children (33.3%) had a diagnosis of Mild Intellectual Disability—MID group (IQ mean = 64.50; SD = 5.42, Min–Max = 51–70); forty had a diagnosis of borderline intellectual functioning—BIF group (IQ mean = 78.23; SD = 3.26, Min–Max = 72–84) and the control group consisted of forty typically developing children (IQ mean = 103.90; SD = 6.50, Min–Max = 90–110). The three levels of intellectual functioning represent the "IQ Group" variable (level 1 = MID; level 2 = BIF; and level 3 = Control).

The three groups matched in terms of sex, and the average age was similar but not completely matched (MID = 12.03; SD = 2.36; BIF = 11.25; SD = 2.28; Control = 11.20; SD = 2.64). The intellectual diagnosis was reported and conducted by the health service independently of this study with tools different from those used in the research.

The ecological sample was selected randomly from several Italian child neuropsychiatry services. The children attended the service for a diagnosis of their cognitive functioning, and during a visit to their parents, the research project was presented, and the informed consent form was provided. First, children with intellectual disabilities were identified based on the outcome of their assessments by the service doctors. Coincidentally, a maximum of 40 children were found for both the MID and BIF groups. To equalize the number, the control group was also limited to 40, and the participants were randomly selected.

Data collection took place at the service premises between February and June 2023, and the participants were met after their parents/guardians submitted signed consent forms. The inclusion criteria for the study were (a) understanding of the Italian language, as Italians or with a prevalent Italian linguistic code as indicated by their respective parents; (b) absence of severe developmental or autism spectrum pathologies; (c) absence of auditory and visual deficits. This study involved an assessment of suggestibility and non-verbal intelligence measures and (d) children with medium and severe intellectual deficits resulting from the assessment carried out by the medical team. To ensure that the

participants had sufficient linguistic skills and that they sufficiently understood the verbal stimuli presented in the study, the cases in which Immediate Recall was zero were excluded.

*2.2. Procedure*

The instruments were administered following the same procedure with all the participants: the Gudjonsson Suggestibility Scale 2 (GSS2) was administered first, while the non-verbal intelligence task was administered during the latency phase between the memory task and the suggestive interview of the GSS2. The tools were administered by two researchers who contributed to the standardization of the GSS2 on the Italian sample (Gudjonsson et al. 2016) and, therefore, have extensive training and experience in administering the GSS2. The administration of the instruments took place individually in a room of the child neuropsychiatry service, thereby ensuring no interference that could alter the administration procedure. Participation was voluntary and without any compensation. The evaluation process carried out by the service included other and different tests from those used in the study. The medical staff, with the authorization of the parents, provided information on the total IQ score of the children to allow a more accurate division of participants into groups. The administration of the Raven Matrices occurred following the medical diagnosis and showed convergence with the score obtained by the medical staff.

The ethics committee of the University of Urbino approved the study with specific recruitment of children. The ethical guidelines of the Declaration of Helsinki were followed and respected in the study. Before proceeding with the recruitment of participants, informed consent signed by the parents was obtained, which contained information on the objective of the study and guarantees anonymity and conservation of sensitive data. The research activity did not include PPE, and the administration took place in healthcare facilities in compliance with workplace safety regulations.

*2.3. Instruments*

2.3.1. Gudjonsson Suggestibility Scales

The Gudjonsson Suggestibility Scale 2 (GSS2; Gudjonsson 1997) is a validated instrument for measuring suggestibility levels in children aged 7 to 16 (Gudjonsson et al. 2016). The Italian GSS2 has acceptable reliability and internal consistency (Cronbach's alpha coefficient: Yield 1, $\alpha = 0.81$; Yield 2, $\alpha = 0.83$; Shift, $\alpha = 0.71$; and Total Suggestibility, $\alpha = 0.77$; Gudjonsson et al. 2016). The Italian version was already used in several studies as well as in other works involving children of different ages and with intellectual disabilities (Vagni et al. 2015, 2017, 2018, 2022; Gudjonsson et al. 2020, 2021, 2022).

The GSS2 involves reading a story that requires Immediate Recall. The Immediate Recall score indicates the number of items correctly recalled (maximum score is 40). From the Immediate Recall, it is possible to obtain the number of memory errors: distortions (small alterations of the original information) and fabrications (added and modified information). The sum of distortions and fabrications constitutes the confabulation score.

The administration of the GSS2 involves a latency phase of approximately 50 min after the collection of the immediate memory. Then, we proceed to administer the suggestive interview consisting of 15 leading questions and 5 neutral questions relating to the previously read story. The suggestive questions are of two types: with a dichotomous answer (yes/no; for example, "Was the boy's bicycle damaged by falling to the ground?") and with a double alternative of wrong answer (such as "Did John grab the boy's arm or shoulder?). Once the interview is over, negative feedback is given ("You made some mistakes. I will repeat the questions. Try to be more precise") by repeating the administration of the same interview. Acceptance responses to the second interview are Yield 2. The number of responses changed from the first to the second interview is the Shift score. The total interrogative suggestibility score (total SI) is given by the sum of Yield 1 and Shift.

Participants' GSS2 responses were also coded as resistant behavioral responses (Gudjonsson et al. 2021, 2022) and concern suggestion-rejecting responses at both interviews. Responses were recorded as follows:

- "NO1" and "NO2" refer to the "no" answers given to Yield 1 and Yield 2, respectively;
- "DE1" and "DE2" are the "Direct Explanation" answers given at both the first and second interviews. Examples of these answers are "this wasn't in the story" or "It wasn't said";
- "DK1" and "DK2" are the "I don't know" or "I don't remember" answers given at Yield 1 and Yield 2, respectively.

### 2.3.2. Raven's Progressive Matrices

This tool measures non-verbal intellectual abilities through the use of abstract and incomplete geometric figures for which one is asked to find the correct sequence with increasing difficulty. Raw scores are converted into centiles and corresponding IQ values by age group; up to 11 years of age, children perform the Colored Progressive Matrices (CPM; Raven 1984; Belacchi et al. 2008) ($\alpha = 0.94$), while adolescents from 12 years of age perform the Standard Progressive Matrices (SPM; Raven 1954) ($KR - 20 = 0.91$)

### *2.4. Analytical Strategy*

A preliminary bivariate Pearson's correlation was conducted between GSS2 scores, age, and IQ. A one-way ANOVA with Tukey post hoc was performed to show IQ-group differences in GSS2 scores.

A G*Power analysis For MANOVA using Pillai's Trace and a priori power analysis was conducted, with 3 groups, 8 dependent variables, $\alpha = 0.05$, power ($1 - \beta$ err. prob.) = 0.95, and medium effect size.

A Multivariate Analysis of Variance (MANOVA) model was conducted assuming the Immediate Recall, memory errors, and GSS2 scores as dependent variables to verify the effects of IQ Group as fixed factor and age as covariates.

A second bivariate Pearson's correlation was made between Resistant Behavioural Responses (RBR), IQ, and age.

A G*Power analysis for repeated measures linear model using Pillai's Trace (effect size F (V) = 0.5 on mean and covariance matrix and for a priori power analysis), with 3 groups, 6 measurements, $\alpha = 0.05$, and power = 0.95 was conducted to determine sample size.

A repeated measures linear model was generated assuming the RBR scores as dependent variables (NO1, NO2, DE1, DE2, DK1, and DK2), IQ Group as fixed factor, and age as covariate. IQ-Group was a variable with three levels: 1 = Mild Intellectual Disability, 2 = borderline intellectual functioning, and 3 = Control.

### 3. Results

#### *3.1. Hypothesis 1: MID and BIF Show More Memory Errors and Immediate Suggestibility Levels than Typically Developing Children*

Pearson correlations were performed to investigate the association between GSS2 scores, age, and IQ (Table 1). IQ showed significant negative correlations with memory errors and suggestibility scores and positive correlations with Immediate Recall. Age showed no significant correlation.

**Table 1.** Correlation coefficients among GSS2 scores, age, and IQ (N = 120).

| Variable | Age | IQ | IR | Total IS |
|---|---|---|---|---|
| IR | −0.103 | 0.744 *** | — | −0.554 *** |
| Distortion | −0.044 | −0.304 ** | −0.279 ** | 0.271 ** |
| Fabrication | 0.091 | −0.329 *** | −0.427 *** | 0.348 *** |
| Confabulation | 0.033 | −0.420 *** | −0.470 *** | 0.411 *** |
| YIELD1 | 0.051 | −0.529 *** | −0.555 *** | 0.880 *** |
| YIELD2 | 0.027 | −0.492 ** | −0.491 *** | 0.733 *** |
| SHIFT | −0.055 | −0.2364 *** | −0.397 *** | 0.856 *** |
| Total IS | 0.050 | −0.517 *** | −0.554 *** | — |

Note. IR = Immediate Recall of the Gudjonsson Suggestibility Scale; Total IS = Total Interrogative Suggestibility. Significance level is marked are follows: ** $p < 0.01$, *** $p < 0.001$.

A one-way ANOVA was generated to detect differences in memory task and suggestibility scores in relation to the different IQ Groups (Table 2). The results showed significant differences in both memory and suggestibility scores. The Tukey post hoc comparisons revealed significant differences between the control group and the other two groups. No difference was detected by Tukey post hoc between Mild Intellectual Disability and borderline intellectual functioning groups. Table 2 shows only the significant post hoc.

**Table 2.** Mean, standard deviations, and Analysis of Variance in IQ-Group on GSS2 scores (N = 120).

| Variable | MID (N = 40) | BIF (N = 40) | Control (N = 40) | F | Post Hoc | d |
|---|---|---|---|---|---|---|
| | Mean (SD) | Mean (SD) | Mean (SD) | | | |
| IR | 6.00 (3.13) | 8.13 (3.80) | 17.68 (5.18) | 90.80 *** | C-MID = 11.68 *** <br> C-BIF = 9.55 *** | 2.72 <br> 2.10 |
| Distortion | 0.53 (0.82) | 0.75 (0.84) | 0.03 (0.16) | 11.84 *** | C-MID = −0.50 ** <br> C-BIF = −0.73 *** | 0.85 <br> 1.19 |
| Fabrication | 0.65 (0.70) | 0.63 (1.039) | 0.04 (0.15) | 9.54 *** | C-MID = −0.63 ** <br> C-BIF = −0.60 ** | 1.21 <br> 0.80 |
| Confabulation | 1.77 (1.11) | 1.38 (1.27) | 0.05 (0.32) | 20.75 *** | C-MID = −1.23 *** <br> C-BIF = −1.33 *** | 2.11 <br> 1.44 |
| Yield 1 | 8.80 (3.47) | 8.50 (3.32) | 3.60 (2.11) | 37.21 *** | C-MID = −5.20 *** <br> C-BIF = −4.90 *** | 3.00 <br> 1.76 |
| Yield 2 | 9.25 (2.67) | 9.30 (3.14) | 4.75 (3.01) | 31.46 *** | C-MID = −4.50 *** <br> C-BIF = −4.50 *** | 1.58 <br> 1.48 |
| Shift | 6.43 (3.86) | 6.73 (3.12) | 3.00 (1.99) | 18.00 *** | C-MID = 3.43 *** <br> C-BIF = −3.72 *** | 1.12 <br> 1.43 |
| Total IS | 15.23 (5.71) | 15.28 (5.30) | 6.60 (3.58) | 40.69 *** | C-MID = −8.63 *** <br> C-BIF = −8.68 *** | 1.81 <br> 1.92 |

Note. MID = Mild Intellectual Disability Group; BIF = borderline intellectual functioning Group; IR = Immediate Recall of the Gudjonsson Suggestibility Scale; Total IS = Total Interrogative Suggestibility; C = control group. Significance level is marked are follows: ** $p < 0.01$; *** $p < 0.001$.

*3.2. Hypothesis 2: Low IQ Levels Lead to Poorer Immediate Recall, a Greater Number of Memory Errors, and Higher Interrogative Suggestibility Scores*

A G*Power analysis for MANOVA using Pillai's Trace and a priori power analysis was conducted, with three groups, six dependent variables, $\alpha = 0.05$, power ($1 - \beta$ err. prob.) = 0.95, and medium effect size (Cohen 1988; Steyn and Ellis 2009). A sample of 93 would identify with critical F = 1.80 and a power of 0.95.

A Multivariate Analysis of Variance (MANOVA) model was performed assuming the Immediate Recall, memory errors, and GSS2 scores as dependent variables to verify the effects of IQ Group as a fixed factor (1 = MID; 2 = BIF; 3 = Control) and age as covariates. Age was included in the model even if Table 1 did not show significant correlations as a control variable. Age normally affects memory tasks and suggestibility on the GSS2 (Gudjonsson et al. 2016), and for this reason, it was included in the model.

The model showed a main effect for IQ_Group (Pillai's trace: Val 0.742; $F_{(12; 224)} = 11.005$; $p < 0.001$; $\eta^2 = 0.371$). Between subject effects for IQ Group were on all variables: Immediate Recall (F = 88.61; $p < 0.001$; $\eta^2 = 0.60$), Distortion (F = 11.92; $p < 0.001$; $\eta^2 = 0.17$), Fabrication (F = 9.16; $p < 0.001$; $\eta^2 = 0.14$), Yield 1 (F = 36.65; $p < 0.001$; $\eta^2 = 0.39$), Yield 2 (F = 31.19; $p < 0.001$; $\eta^2 = 0.35$), Shift (F = 18.47; $p < 0.001$; $\eta^2 = 0.24$). The parameter estimates relating to the three groups are similar to those reported in Table 2. No effect was obtained for age.

*3.3. Hypothesis 3: Lower IQ Levels Reduce Resistant Responses with Higher Source Monitoring Capacity, Such as "NO" and "Direct Explanation" Responses*

Preliminary Pearson's bivariate correlations were carried out between RBRs, IQ, and age (Table 3).

**Table 3.** Correlation coefficients among RBR scores, age, and IQ (N = 120).

| Variable | Age | IQ | IR | Total IS |
|---|---|---|---|---|
| NO1 | −0.275 ** | 0.208 * | 0.317 *** | 0.550 *** |
| DE1 | 0.128 | 0.460 *** | 0.375 *** | −0.513 *** |
| DK1 | 0.220 * | 0.072 | 0.059 | −0.147 |
| NO2 | 0.139 | 0.170 | 0.194 * | −0.390 *** |
| DE2 | 0.053 | 0.407 *** | 0.373 *** | −0.470 *** |
| DK2 | 0.103 | 0.067 | 0.112 | −0.158 *** |

Note. IR = Immediate Recall of the Gudjonsson Suggestibility Scale; Total IS = Total Interrogative Suggestibility. DE1 and DE2 = Direct Explanation for Yield 1 and Yield 2, respectively; DK1 and DK2 = Don't Know for Yield 1 and Yield 2, respectively. Significance level is marked are follows: * $p < 0.05$, ** $p < 0.01$, *** $p < 0.001$.

The differences in RBR scores for the three groups were reported in the Appendix A (see Table A1).

A G*Power analysis was conducted to verify the necessary sample size to follow a repeated measures analysis.

A repeated measures linear model using Pillai's Trace (effect size F (V) = 0.75 on mean and covariance matrix and for a priori power analysis), with three groups, six measurements, $\alpha = 0.05$, and power = 0.95, was performed, and a sample of 27 was used for identification.

To verify the IQ effect on resistant responses to both the first and second interviews, which occurred after the negative feedback, a repeated measures linear model was generated. The resistant responses (NO, Direct Explanation, and Don't Know responses) in both interviews (Yield 1 and Yield 2) were assumed as dependent variables, IQ Group with three levels (MID, BIF, and Control) as a fixed factor and age as a covariate.

The model showed significant main effects for RBRs (Pillai's trace: Val 0.327; $F_{(5, 112)} = 10.899$; $p < 0.001$; $\eta^2 = 0.33$), RBR*IQ Group (Pillai's trace: Val 0.266 $F_{(10, 226)} = 3.464$; $p < 0.001$; $\eta^2 = 0.13$), and RBR*age (Pillai's trace: Val 0.146 $F_{(5, 112)} = 3.814$; $p < 0.01$; $\eta^2 = 0.15$).

Between subject effects was for IQ Group (F = 38.887; $p < 0.001$; $\eta^2 = 0.40$) with significant parameter estimates on NO1 ($t_{(Control-MID)} = 2.71$; $p < 0.01$; $d = 0.64$; $\eta^2 = 0.06$; C.I. 95%: 0.46–3.69; and $t_{(Control-BIF)} = 3.39$; $p < 0.01$; $d = 0.77$; $\eta^2 = 0.09$; C.I. 95%: 0.64–3.86), NO2 ($t_{(Control-MID)} = 2.44$; $p < 0.05$; $d = 0.81$; $\eta^2 = 0.05$; C.I. 95%: 0.16–3.09; and $t_{(Control-BIF)} = 2.88$; $p < 0.01$; $d = 0.58$; $\eta^2 = 0.07$; C.I. 95%: 0.31–3.24), DE1 ($t_{(Control-MID)} = 5.66$; $p < 0.001$; $d = 1.06$; $\eta^2 = 0.22$; C.I. 95%: 1.65–4.30; and $t_{(Control-BIF)} = 4.65$; $p < 0.001$; $d = 0.85$; $\eta^2 = 0.16$; C.I. 95%: 1.11–3.88), and DE2 ($t_{(Control-MID)} = 4.69$; $p < 0.001$; $d = 0.91$; $\eta^2 = 0.16$; C.I. 95%: 1.30–4.10; and $t_{(Control-BIF)} = 4.29$; $p < 0.001$; $d = 0.81$; $\eta^2 = 0.14$; C.I. 95%: 1.12–3.93).

Between subject effects were also significant for age (F = 15.092; $p < 0.05$; $\eta^2 = 0.11$), but the parameter estimates were significant only for the responses to the first interview: NO1 ($t = -3.00$; $p < 0.01$; $\eta^2 = 0.07$); DE1 ($t = 2.22$; $p < 0.05$; $\eta^2 = 0.04$), and DK1 ($t = 2.55$; $p < 0.05$; $\eta^2 = 0.02$).

## 4. Discussion

The aims of the present study were to verify how children with intellectual disabilities manage a suggestive interview and whether the presence of cognitive deficits limited resistant responses. The results highlighted how IQ has a significant negative correlation with both memory tasks and suggestibility scores (Table 1). In particular, high intellectual abilities led to the production of fewer memory errors, such as distortions and fabrications, and to greater resistance to the factors of interrogative suggestibility.

Children with mild intellectual and borderline cognitive disabilities showed similar performance both in the Immediate Recall task and in suggestive interviews. They also

showed the same tendency to produce memory errors which was significantly higher than the control group (Table 2). The ability to recover information learned even immediately is significantly lower in children with MID and BIF, who may resort to processes of invention and confabulation to cover memory gaps. These results seem to be in agreement with other studies that have highlighted deficits in executive functions, attention, and working memory in these children (Carrasco et al. 2005; Henry 2001; Alloway 2010; Lee and Kim 2022).

However, it is important to highlight how the memory task required by GSS2 is learning and not autobiographical memory. No prediction or inference can be made between Immediate Recall and their memory error production at the GSS2 and their autobiographical capacity.

Children with mild intellectual disabilities and borderline intellectual functioning showed greater suggestive vulnerability than children developing typically. The effect of age, which generally affects suggestibility, seems to be absorbed by the effect of IQ. This can be explained by the fact that due to their cognitive deficit and delay their performance is not weighted according to their chronological age. The impact of IQ on immediate suggestibility in our study confirmed what was highlighted by Gudjonsson (2003) and Roma et al. (2011), who found a significant impact, especially on Yield 1 scores (Gudjonsson 2003).

Children with mild intellectual disabilities and borderline intellectual functioning also showed a vulnerability to the effect of question repetitions after negative feedback. Their Shift and Yield 2 scores were, in fact, significantly higher. These results seem to suggest that intellectual deficits negatively impact the management of external expectations, the sense of self-efficacy, and security in one's memory. These results are, therefore, in agreement with Gudjonsson and Young (2021) in a sample of adults with ID. Having low cognitive abilities to recover and reorganize memory information can lead children with intellectual disabilities, especially to greater trust in the adult interviewer. Distrust memory and interpersonal trust are two factors underlying interrogative suggestibility (Gudjonsson 2003). Lower than average IQs tend to be associated with poor source monitoring skills, which allow the correct original information to be recovered during the recall phase and to recognize one's own recall errors. In fact, low IQs, as highlighted by the results of the study, tend to favor the production of memory errors such as distortions and fabrications. The source monitoring capabilities also allow for the recognition of the misleading information contained in the questions during the interview and the activation of a response of refusal or resistance. Previous studies indicated that resistant responses to leading questions could be of different types in relation to the different degrees of source monitoring (Gudjonsson et al. 2021, 2022; Gudjonsson and Young 2021; Vagni et al. 2023).

Considering the incidence of IQ at all suggestibility scores, its impact on the ability to produce resistant responses, which generally require more mature and complex cognitive abilities, was also verified. According to Gudjonsson et al. (2021, 2022), IQ showed significant positive correlations with NO responses at the first interview (Yield 1) and with Direct Explanation responses at both interviews (Yield 1 and 2).

Age showed a negative correlation with NO answers at the first interview (Yield 1) because, usually, older children are able to provide more accurate answers in terms of source monitoring or admitting that they do not possess the requested information (DK responses) (see Table 3). The repeated measures model demonstrated how age seems to have an impact only on the responses to the first interview (Yield 1). This seems to suggest that following the negative feedback, the cognitive and social skills required to cope with the second suggestive interview and the stress levels go beyond mere age. Children with greater intellectual abilities and typical intellectual functioning tend to have more personal confidence, to use more functional response strategies, and to be able to retrieve information from memory with greater self-efficacy. This may lead them to maintain a good number of "NO" and "Direct Explanation" responses, unlike MID and BIF children.

Children with MID and BIF showed a reduced capacity for source monitoring, and this may have led them to greater vulnerability. Source monitoring is, in fact, a cognitive

ability that allows you to discriminate the source of information (Rossi-Arnaud et al. 2023). Younger children generally show poor source monitoring, and this leads them to accept more misleading information. Children with low intellectual abilities may show difficulty in recognizing whether the information requested is the same as that held in memory and in rejecting misleading information more decisively.

According to Gudjonsson and Young (2021), the main characteristic of the DE response style is that it involves effective source monitoring, a socio-cognitive mechanism associated with confidence in one's memory, critical strategic thinking, and self-efficacy. The results of the MANOVA and repeated measure models demonstrated how MID and BIF children not only show difficulties in cognitive tasks (such as attention, memory errors, and source monitoring) but also in managing the more social aspects of suggestive interaction. In fact, the effects of IQ also showed similarly significant results in the second suggestive interview (Yield 2) after the negative feedback, and this seems to suggest how feelings of low self-esteem and self-efficacy could increase trust and adherence to the interviewer. However, it is not possible to establish whether this occurs in a fully conscious manner.

IQ showed no impact on Don't Know responses, which instead seems to depend on age. According to Waterman and Blades (2011), answering "I don't know" is a resistant behavior that allows you not to immediately accept the suggestion. People tend to give "I don't know" answers when they are actually uncertain or when they cannot recall the information at that moment (Scoboria et al. 2008). Ceci and Bruck (1993) found that the ability to answer "I don't know" to a question that does not provide the suggested answer or response alternative leads to more confident and mature cognitive and expressive skills. IQ had no impact on favoring or reducing the production of "I don't know" answers, while the impact was detected in relation to the age variable.

The ability to resist and manage the socio-emotional pressures linked to negative feedback is only partly due to cognitive abilities, as even the control children show vulnerability, and the increase in age has no impact on favoring resistant responses to the second interview.

In fact, all the children, even those in the control group, showed an increase in suggestive vulnerability after the negative feedback. Age seems to have allowed a greater capacity for resistance only to the first interview, which mainly refers to the cognitive factors involved in the suggestive interview (Gudjonsson 2003). In other words, age does not appear to have had a protective effect on socio-emotional pressure. The ability to manage negative feedback and the socio-emotional pressure of a suggestive interview does not appear to be age-related when groups are homogeneous and when children are over 12 years old (Gudjonsson 2003; Maiorano and Vagni 2020; Vagni et al. 2023). Probably, the ability to give "I don't know" answers refers to linguistic skills that were not included in this study. Furthermore, intellectual difficulties can reduce attention skills and working memory as well as comprehension and language production, and these skills could affect their performance on the GSS2 (Carrasco et al. 2005). Comprehension and expressive difficulties could make children more vulnerable to changing their responses following negative criticism (Melinder et al. 2005).

Finally, the similar vulnerability and limitations in providing resistant responses in MID and BIF children seem to indicate that the cause is not directly due to the degree of intellectual delay but to cognitive functioning for which some abilities are limited. In fact, BIF children do not have intellectual delay but have limited functioning in some cognitive functions, such as those responsible for ensuring correct source monitoring (in particular, attention and working memory may be limited or discontinuous). Therefore, as there are no differences in the results between MID and BIF, their high suggestibility and low ability to provide resistant responses seem to be due to deficits in source monitoring. Other cognitive variables could also be involved, which, however, were not taken into consideration in this study and which could guide future investigations.

The study presents some limitations related first to the sample size. The three groups, although homogeneous in age and gender, are small in size, and this limits the general-

ization of the results. The sample does not include child witnesses, and this may limit the full extension of the results to the forensic context. In fact, previous studies found that child witnesses tend to present a greater vulnerability to negative criticism compared to children in the control group (Gudjonsson et al. 2021, 2022; Vagni et al. 2017, 2018). However, compared to Yield 1, in the absence of other clinical conditions such as anxiety or post-traumatic disorder, significant differences do not always occur between witnesses and controls (Vagni et al. 2018; Gudjonsson 2018).

Furthermore, the study took into consideration a non-verbal measure of intelligence. In responding to a suggestive interview, it is likely that other cognitive abilities, such as attention, semantic memory, and language, could also be involved and that Raven's matrices do not measure. In fact, in children with intellectual difficulties, even comprehension and linguistic production skills may affect their performance on the GSS2 (Melinder et al. 2005).

Finally, the study measured immediate suggestibility but not delayed suggestibility, that is, the tendency to modify the original memory as a result of misinformation. For future studies, it would be useful to also administer an instrument for linguistic skills to be related to the ability to give more cognitively complex answers.

## 5. Conclusions

In the forensic field, it is common to listen to children with cognitive disabilities because they are victims or witnesses of crimes. Their performance is often inferior in terms of memory compared to children with normal abilities. Often, to overcome their difficulties, direct and suggestive questions are asked, or greater pressure is created in relation to their low performance. These factors cause children with intellectual disabilities to make more memory errors, to give in to suggestive questions, and to provide less resistant answers compared to suggestibility factors. The study confirmed how their ability to resist leading questions is affected by their intellectual limitations rather than across the various ages. In fact, as these children grow, they continue to show their difficulties and have a lower performance.

The results of the study highlight how, in the forensic field, the testimony expert must carry out a careful analysis of their ability to source monitor and provide resistant answers. The evaluations of testimonial reliability and efficiency of children with MID are often based on stereotypes rather than objective parameters. Borderline intellectual functioning children who do not have cognitive disabilities present a suggestive vulnerability similar to that of children with delay.

In cases of legal listening to children with cognitive difficulties, it is important to explain the Ground Rule, formulate questions in a simpler way, and avoid exposing them to a sense of their inability to answer the questions.

The forensic interview should also take into consideration that children with cognitive difficulties may also have limitations in attention and language functions. This tends to suggest that questions should be short and simple. The interview should be free of elements that reinforce criticism of performance or uncertainty about what was said, as ID children appear to be particularly vulnerable to social pressure factors regardless of their age. In consideration of these intellectual impairments, it appears necessary to highlight how the expert who helps the judge in collecting the testimony must have specific training and know their limits to allow the formulation of questions that can help to recover the greatest amount of information, formulate short and simple questions, avoid increasing a sense of ineffectiveness in them.

Finally, considering the fact that intellectual impairment also leads to language difficulties, it seems useful to also include a measure of children's linguistic skills in future research. Furthermore, to ensure the effective extensibility of the results in the forensic field, future research should include the recruitment of child witnesses.

**Author Contributions:** Conceptualization, M.V. and V.G.; methodology, M.V. and V.G.; validation, M.V. and V.G.; formal analysis and data curation, M.V.; writing—original draft preparation, M.V. and V.G.; writing—review and editing, M.V. and V.G. All authors have read and agreed to the published version of the manuscript.

**Funding:** This research received no external funding.

**Institutional Review Board Statement:** This study was conducted according to the guidelines of the Declaration of Helsinki and approved by the Ethics Committee of University of Urbino on 18 March 2020 (Comitato Etico per la Sperimentazione Umana—CESU of the University of Urbino).

**Informed Consent Statement:** Informed consent was obtained from all subjects involved in the study.

**Data Availability Statement:** The data presented in this study are available upon request from the corresponding author.

**Conflicts of Interest:** The authors declare no conflict of interest.

## Appendix A

**Table A1.** Difference to RBRs in the MID, BIF, and control groups (N = 120).

| Variable | MID (N = 40) | BIF (N = 40) | Control (N = 40) | F |
|---|---|---|---|---|
| | Mean (SD) | Mean (SD) | Mean (SD) | |
| NO1 | 5.30 (3.24) | 5.13 (2.60) | 7.38 (3.23) | 6.77 ** |
| NO2 | 4.18 (2.50) | 4.63 (3.02) | 6.30 (2.72) | 5.11 ** |
| DE1 | 0.33 (0.83) | 0.75 (1.79) | 3.30 (3.86) | 16.54 *** |
| DE2 | 0.55 (1.32) | 0.73 (1.83) | 3.25 (3.99) | 13.07 *** |
| DK1 | 0.38 (1.34) | 0.53 (1.30) | 0.88 (1.42) | 2.11 * |
| DK2 | 0.53 (1.06) | 0.40 (1.01) | 0.64 (1.44) | 0.36 |

Note. MID = Mild Intellectual Disability Group; BIF = borderline intellectual functioning Group; DE1 and DE2 = Direct Explanation for Yield 1 and Yield 2, respectively; DK1 and DK2 = Don't Know for Yield 1 and Yield 2, respectively. Significance level is marked are follows: * $p < 0.05$, ** $p < 0.01$; *** $p < 0.001$.

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
