# Peer review of "Interrogative Suggestibility and Ability to Give Resistant Responses in Children with Mild Intellectual Disabilities and Borderline Intellectual Functioning"

_socsci, doi:10.3390/socsci13020077_

Round 1
Reviewer 1 Report
Comments and Suggestions for Authors
1. What is the main question addressed by the research?
The suggestibility of children with mild mental retardation can be a source of difficulties not only in the child's life but also in adulthood. Especially in cases where inclusive education is concerned, the level of risk is higher. And it is these children who are often recommended to the educational environment of mainstream schools. The work aimed to describe the relationship between intellect and suggestibility on a selected sample of respondents. In both cases, the instrument used was a standardized test.
2. Do you consider the topic original or relevant in the field? Does it address a specific gap in the field?
The elaborated topic is a high-quality contribution to the knowledge of the monitored relationship in the compared items.
3. What does it add to the subject area compared with other published material?
In this study, the "borderline" intelligence band is compared with the results of the GSS2 test. This adds work on originality.
4. What specific improvements should the authors consider regarding the
methodology? What further controls should be considered?
The methodology is processed according to common standards, only the hypotheses could be better formulated to describe the search for a relationship between variables. More "school" type.
5. Are the conclusions consistent with the evidence and arguments presented
and do they address the main question posed?
The conclusions of the authors' work are entirely within the text's intentions.
6. Are the references appropriate?
The sources used and cited are formally correct and content-wise appropriate for the topic of the article.
7. Please include any additional comments on the tables and figures.
The tables are clearly arranged and the results are presented in a standard way.
Comment:
It would still be useful to complete the text by referring to some works that show the specific competence requirements of psychologists working with and preparing persons with mental deficiencies.
Mild mental retardation can cause deficits in language competence. Please describe how you investigated " The inclusion criteria for the study were: a) understanding of the Italian language;. Evidence with already published studies.
Author Response
Comment:
It would still be useful to complete the text by referring to some works that show the specific competence requirements of psychologists working with and preparing persons with mental deficiencies.
Reply: Really thanks to the reviewer for this suggestion. We have added some considerations on the expert's competence and some indications on how the testimony of these children should be collected in the conclusions of our article
Mild mental retardation can cause deficits in language competence. Please describe how you investigated " The inclusion criteria for the study were: a) understanding of the Italian language;. Evidence with already published studies.
Reply: The manuscript has been modified in its various parts. Some concepts have been clarified with greater reference support. The procedure and inclusion/exclusion criteria sections have also been clarified and improved. We agree with the reviewer that in MID and BIF children language can become highly relevant. We have included also this suggestion in the conclusions of the article and as a possible future aim of future studies. The manuscript was also reviewed by an expert in English language.
Reviewer 2 Report
Comments and Suggestions for Authors
See attached file

They are acceptable
Author Response
Review of socsci-2745825:Interrogative suggestibility and ability to give resistant responses in children with mild intellectual disabilities and borderline intellectual functioning This is a straightforward study. You report associations for both memory and suggestibility of various types with intellectual ability. The study is easy to follow as you employed a well-used procedure for analysing these constructs. In the main the data are clear and what would be expected. There are a few omissions (e.g., an under-specification of which ‘liner model ‘was used in the second set of analyses, a lack of effect sizes for the t tests), but otherwise all is clear.
Reply; thank you to the reviewer for the comments and suggestions. The entire text has been revised in its missing parts also following the indications of the other reviewers. Effect sizes for the t tests were added. The text was also revised by an English language expert.
However, I do have two reservations about the paper:
- I am a little confused by the stated rationale for your study. You write (page 2, lines 77-78): ‘In the forensic context, it is likely that child will be subjected to leading questions. The main purpose of this study is to verify if intellectual disability may lead children to inaccurate recall and greater suggestive vulnerability.’ If children are asked misleading questions doesn’t this invalidate the procedure as a whole? Are you assuming that a child with above average intellectual functioning will see through misleading questions, while a child with BIF or below will be susceptible? If not, I feel that you need a stronger justification about the relationship between the type of misleading question posed and the child’s response, mediated by the extent of the child’s intellectual problems? If so, I am not sure about the validity of this exercise, as we know that children with intellectual impairment are susceptible to such questions. Perhaps you are interested in questions related to the category BIF but, as my next point suggests, you do not make this very clear.
- I am not 100% clear about why you focus on BIF as opposed to the four categories of intellectual disability that have traditionally been used. I realise that the categorical system and one based on SDs are constitutive on one another, but the categorical system has been used more. On page 1 you state that the BIF has received ‘growing attention’ or ‘commonly defined as….’ and then, ironically cite no paper to support your case. So I am left wondering what the benefits of using BIF might be and even the suspicion that your statistics worked only when this system was used. On page 2, you refer to a general review on BIF, but little detail. You then cite a review (page 4, para 1) which summarises the literature (accurately to my reading of it) as identifying that moderate intellectual ability is associated with problems of suggestibility, so why focus on mild ID and BIF? Finally, having sampled these groups some of your analyses use IQ in linear analyses, so why bother here with these groupings or, more particularly, this range of children based upon their general IQ scores. The Discussion seems sensible, although I wonder whether anything you report here adds to the existing literature, as all the points have been made before. Without an analysis of point 2 above that is convincing I remain unconvinced that your paper makes a contribution.
Reply for both points: We agree with some considerations made by the reviewer. However, we believe it is necessary to specify some aspects to avoid misunderstandings about the objectives of the study. Most studies involving participants with intellectual impairment actually involved children or adults with ID. Few studies have involved BIF children with respect to their suggestibility. BIF is a clinical condition recognized by the DSM 5TR but not strictly defined as cognitive delay, although their cognitive functioning is lower in some skills than typically developing children. This clinical category has been used in several studies, but the literature is lacking in demonstrating the functioning of BIF children with respect to interrogative suggestibility factors.
We tried to specify this aspect in the introduction of the article, also citing some studies. The reviewer 1 wrote: “The elaborated topic is a high-quality contribution to the knowledge of the monitored relationship in the compared items. In this study, the "borderline" intelligence band is compared with the results of the GSS2 test. This adds work on originality“.
Reviewer 4 highlighted that: “I thoroughly enjoyed reading this paper and believe it contributes to existing knowledge in this field”.
The study attempted to verify the existence of any differences between the three groups: MID, BIF and typically developing. Since studies on BIF children are few, the present study aimed to highlight their functioning with respect to leading questions, highlighting how they appear similar to MID children.
However, the main aspect of originality is given by the application of the resistant response model in children with different degrees of delay. The RBR model has never been applied to MID and BIF children so far.
Unfortunately in forensic practice suggestive questions are used with children without invalidating their testimony. In other words, despite good practices advising against the use of suggestive questions, they are used. Typically developing children also show a vulnerability to suggestibility factors, although often - but not always - it is lower than that of children with delayed impairment. More than the number of suggestions accepted, it appears important to understand what type of answers they are able to give in relation to their source monitoring capacity. We believe that the latter is precisely affected by intellectual abilities. The study has several limitations that we have tried to highlight.
We hope to have clarified some aspects of our study
Minor points
I’m not sure that you should write 1.5 line paragraphs. There are a few in this ms.
Reply: We followed the journal's rules and the editor assured us that in case of errors, all manuscripts are checked by the editorial staff before publication
Line 46: Shouldn’t ‘tipically’ be ‘typically’?
Reply: we apologize for the typo. The manuscript was also reviewed by an expert in English
Line 219: You report alphas of .77 and even .71 as being good, but this conflicts with Cronbach’s own interpretation of this statistic.
Reply: we replaced good with acceptable.
Table 1 and subsequent tables: I trust that journal policy is for 3 dp, rather than the 2 recommended, e.g., by the APA? Two dp might make it easier to read your tables.
Reply: we report what is written above after consulting the associate editor
Reviewer 3 Report
Comments and Suggestions for Authors
Researchers examined immediate suggestibility in children and found that children who have been diagnosed with Mild Intellectual Impairment and Borderline Intellectual impairment exhibit similar levels of suggestibility, both of which are higher than the control group. Researchers assessed immediate suggestibility using the Gudjonsson Suggestibility Scale-2 and specifically examined resistant behavioral responses to unanswerable questions during the interview. Results revealed that IQ scores (assessed via Raven’s matrices) were positively correlated with appropriate responses to unanswerable questions, which was illustrated by children responding with “no” or providing “Direct Explanations” indicating the question was unanswerable.
The authors are assessing a critical question in an attempt to understand how children with intellectual impairment are likely to perform during a suggestive interview. However, there are several issues in the manuscript that should be addressed to clarify and strengthen the contribution of this study to the literature. A major concern is the overall quality of the paper that is in critical need of polishing and refining. It is also critical to provide additional information about the methods in order to assess methodological rigor (addressed below) and improve the interpretation of hypothesis 3, which the authors highlight is the novel contribution of their study. Additional feedback on each section is provided below:
Intro
-
The discussion of the literature is not linear and can benefit from restructuring, I suggest first introducing the effects of impairment on suggestibility and then provide results for MID and then BIF (highlighting that there is little research on BIF)
-
Line 105 clarify what “development process” is referring to and how it relates to IS
-
Line 109 - introduction of socio-emotional factors seems tangential to the topic since it is not assessed in this study
-
Line 112 - can benefit from clear discussion/introduction of the types of responses being examined. Authors introduce findings for “accepting” the suggested response but then focus on “resisting” behavioral responses
-
Line 130 what does most stable mean when referring to DE
-
Line 142 GSS2 has not been introduced to interpret yield 1 nomenclatures
-
Authors introduce Klemfuss and Olaguez review that finds inconsistencies in the literature and then summarize inconsistent findings - would help to extend the conclusions of Klemfuss and Olaguez to highlight the methodological differences rather than restating that the review has already concluded
-
Comparing correlations between IQ and suggestibility vs comparing average suggestibility based on group assignment (Mild ID vs control)
-
Comparing # of distortions vs overall suggestibility score (studies are comparing different DVs)
-
Present study
-
Hypotheses 2 and 3 should be directional (currently uses “affects”)
-
Hypothesis 3: social cognitive mechanism associated with confidence and trust in one’s memory decreasing resistant responses
-
Findings for this hypothesis (measures) are not reported
Method
-
Sample selection
-
What is the sampling frame? Where were participants “found” and how did they recruit them to participate? What was participant compensation?
-
Participant compensation
-
Who conducted the sessions - where were they administered
-
Did ravens score match the pre existing diagnoses? Given that researchers themselves did not assess IQ diagnosis, it is not clear whether the results from the current study matched the existing diagnosis - critical to match to previous research and determine the validity of prev diagnoses
-
Analytical strategy should match hypotheses
Results
-
Turkey is use throughout and should be tukey
-
Given that some variables are composites, including both the individual variable and the composite is duplicating the results and inflating the r value
-
There are inconsistencies when referring to the conditions (not using MID or BIF throughout)
-
Hypothesis 2
-
Should be directional and focused on comparison not causal
-
It is likely there is not enough power for a MANOVA
-
the covariate is also reported as a Main effect?
-
What are the post hoc findings for IQ main effects?
-
Hypo 3
-
Same comment as hyp 2
-
Confidence and trust are not assessed
-
Clarify Line 318 (IQ group with 3 levels)
-
Numbers would be easier to follow in a table
-
Is this a big enough sample to power a GLM?
-
These results need additional interpretation/explanation given the complexity of the model
Comments on the Quality of English Language
The paper is in critical need of major revisions to address sentence structure and typographical errors. There were too many issues to highlight in the comments.
Author Response
Reviewer 3
Researchers examined immediate suggestibility in children and found that children who have been diagnosed with Mild Intellectual Impairment and Borderline Intellectual impairment exhibit similar levels of suggestibility, both of which are higher than the control group. Researchers assessed immediate suggestibility using the Gudjonsson Suggestibility Scale-2 and specifically examined resistant behavioral responses to unanswerable questions during the interview. Results revealed that IQ scores (assessed via Raven’s matrices) were positively correlated with appropriate responses to unanswerable questions, which was illustrated by children responding with “no” or providing “Direct Explanations” indicating the question was unanswerable.
The authors are assessing a critical question in an attempt to understand how children with intellectual impairment are likely to perform during a suggestive interview. However, there are several issues in the manuscript that should be addressed to clarify and strengthen the contribution of this study to the literature. A major concern is the overall quality of the paper that is in critical need of polishing and refining. It is also critical to provide additional information about the methods in order to assess methodological rigor (addressed below) and improve the interpretation of hypothesis 3, which the authors highlight is the novel contribution of their study. Additional feedback on each section is provided below:
Reply: We thank the reviewer for all the suggestions given, allowing us to make the manuscript clearer and to make the introduction and examination of the results more complete. We greatly appreciated the indications which we fully shared by making the relevant changes. The manuscript was also reviewed by an expert in English language.
- Intro
- The discussion of the literature is not linear and can benefit from restructuring, I suggest first introducing the effects of impairment on suggestibility and then provide results for MID and then BIF (highlighting that there is little research on BIF)
- Line 105 clarify what “development process” is referring to and how it relates to IS
- Line 109 - introduction of socio-emotional factors seems tangential to the topic since it is not assessed in this study
- Line 112 - can benefit from clear discussion/introduction of the types of responses being examined. Authors introduce findings for “accepting” the suggested response but then focus on “resisting” behavioral responses
- Line 130 what does most stable mean when referring to DE
- Line 142 GSS2 has not been introduced to interpret yield 1 nomenclatures
- Authors introduce Klemfuss and Olaguez review that finds inconsistencies in the literature and then summarize inconsistent findings - would help to extend the conclusions of Klemfuss and Olaguez to highlight the methodological differences rather than restating that the review has already concluded
- Comparing correlations between IQ and suggestibility vs comparing average suggestibility based on group assignment (Mild ID vs control)
- Comparing # of distortions vs overall suggestibility score (studies are comparing different DVs)
Reply: We summarize here the responses to all the suggestions relating to the introduction. the introduction has been significantly restructured and we have inserted some parts that were missing and clarified some sentences and concepts. Let's hope the text is clearer now. We also examined the literature again to clarify some terms and to highlight what has already been examined by previous studies, also highlighting what is so far missing in the panorama of research in this area.
- Present study
- Hypotheses 2 and 3 should be directional (currently uses “affects”)
- Hypothesis 3: social cognitive mechanism associated with confidence and trust in one’s memory decreasing resistant responses
- Findings for this hypothesis (measures) are not reported
Reply: hypotheses 2 and 3 have been reformulated more clearly and with directionality of the expected results
3 Method
- Sample selection
- What is the sampling frame? Where were participants “found” and how did they recruit them to participate? What was participant compensation?
- Participant compensation
- Who conducted the sessions - where were they administered
- Did ravens score match the pre existing diagnoses? Given that researchers themselves did not assess IQ diagnosis, it is not clear whether the results from the current study matched the existing diagnosis - critical to match to previous research and determine the validity of prev diagnoses
Reply: We thank the reviewer for the helpful suggestions. The highlighted information has been reported in the "Method" section to better explain the procedure and criteria for carrying out the study
- Analytical strategy should match hypotheses
Reply: The section was checked to ensure the matched hypothesis and results section. G*Power analyzes have been included
- Results
- Turkey is use throughout and should be tukey
Reply: we are sorry for the mistake
- Given that some variables are composites, including both the individual variable and the composite is duplicating the results and inflating the r value
Reply: We agree with the reviewer and a new MANOVA model has been generated excluding the confabulations which are the sum of distortions and fabrications and total suggestibility which is given by the sum of Yield1 and shift
- There are inconsistencies when referring to the conditions (not using MID or BIF throughout)
Reply: Throughout the text attention has been given to referring more specifically and correspondingly to the MID and BIF condition
4a) Hypothesis 2
- Should be directional and focused on comparison not causal
- It is likely there is not enough power for a MANOVA
- the covariate is also reported as a Main effect?
- What are the post hoc findings for IQ main effects?
Reply: We agree with the reviewer that the sample was limited. We conducted a preliminary G*Power analysis. The authors had already completed it but not reported in the manuscript. The results of the G*Power have been included in the section relating to the second hypothesis.
Post hoc results were not reported in the MANOVA analysis because they were already reported in Table 2 for the one-way Anova. The post hoc comparisons are in line with each other and the authors have chosen not to report redundant results. A new MANOVA model was generated with 6 dependent variables and not 8. The results were reported more clearly. Hypothesis 2 was reformulated.
4b) Hypo 3
- Same comment as hyp 2
- Confidence and trust are not assessed
- Clarify Line 318 (IQ group with 3 levels)
- Numbers would be easier to follow in a table
- Is this a big enough sample to power a GLM?
- These results need additional interpretation/explanation given the complexity of the model
Reply: A G*Power analysis for repeated measures was conducted to verify the necessary sample size. The results have been included in the text. The dependent variables used and the 3 groups linked to IQ have been better specified. The results obtained regarding resistant responses have been better explained in the discussion section. We have entered the C.I. values. for comparisons with significant results
Reviewer 4 Report
Comments and Suggestions for Authors
Dear author(s),
I thoroughly enjoyed reading this paper and believe it contributes to existing knowledge in this field. My suggestions below are provided in order to enhance the quality of the paper overall:
Page 3, lines 127-130 - could the author(s) provide more clarity here as to the point they are trying to make?
Hypotheses: whilst the hypotheses are clear, they do not make reference to the RBR model or type of response although this is mentioned in the results. Could the author(s) be more explicit here to reflect this?
Clearance to work with children: I appreciate that this study was conducted in Italy (certainly the participants were all Italian) but could the author(s) state what security clearance they had or was required to work with children in this manner?
Participant numbers: why was 120 participants used? What was the rationale for this? Did the author(s) conduct a G Power analysis for example? This is raised as a limitation later on in the paper so some clarity around the participant size would be helpful.
Limitations: The author(s) raise some insightful limitations but could they consider the issue of ecological validity? My understanding of the method is that the participants were administered the GSS2, some other measures and then interviewed twice. Can findings be generalised to child witnesses when the methodology did not wholly reflect real-life, e.g., the participant did not 'witness' an event and then was interviewed. This does not detract away from the quality of the paper but I think it would be helpful to include this point in your limitations to demonstrate awareness.
I hope these comments are helpful and assist in enhancing your paper.
Comments on the Quality of English Language
Generally, the quality of English is very good although the paper would benefit from a thorough proof-read, particularly the first half of the paper where there are number of grammatical/spelling errors. Improving these will help with clarity.
Author Response
Dear author(s),
I thoroughly enjoyed reading this paper and believe it contributes to existing knowledge in this field.
Reply: We really thanks to the reviewer for the appreciation and for the useful advice that we tried to follow as best as possible
My suggestions below are provided in order to enhance the quality of the paper overall:
Page 3, lines 127-130 - could the author(s) provide more clarity here as to the point they are trying to make?
Reply: The introduction has also been restructured to follow the suggestions of other reviewers. More attention was given to explaining the theoretical constructs and variables used in the study. The text was reviewed by an English language expert to ensure greater clarity.
Hypotheses: whilst the hypotheses are clear, they do not make reference to the RBR model or type of response although this is mentioned in the results. Could the author(s) be more explicit here to reflect this?
Reply: hypotheses 2 and 3 have been reformulated more clearly indicating the directionality of the expected results
Clearance to work with children: I appreciate that this study was conducted in Italy (certainly the participants were all Italian) but could the author(s) state what security clearance they had or was required to work with children in this manner?
Reply: The ethics committee of our university authorized the research with children by preparing a consent form in which the methods of conduct were described. The research was conducted in the offices of the child neuropsychiatry services of the various hospitals and the methods were carried out according to the rules and regulations of safety at work. Administration of the instruments did not require PPE. The premises, being those of hospitals, complied with the law.
Participant numbers: why was 120 participants used? What was the rationale for this? Did the author(s) conduct a G Power analysis for example? This is raised as a limitation later on in the paper so some clarity around the participant size would be helpful.
Reply: We agree with the reviewer that the sample was limited. We conducted a preliminary G*Power analysis. The authors had already completed it but not reported in the manuscript.
G*Power results were reported both for the MANOVA and repeated measures linear model.
Limitations: The author(s) raise some insightful limitations but could they consider the issue of ecological validity? My understanding of the method is that the participants were administered the GSS2, some other measures and then interviewed twice. Can findings be generalised to child witnesses when the methodology did not wholly reflect real-life, e.g., the participant did not 'witness' an event and then was interviewed. This does not detract away from the quality of the paper but I think it would be helpful to include this point in your limitations to demonstrate awareness.
Reply: Thank you to the reviewer for this suggestion which seems very important to us in order the generalization of the results in the forensic field. Previous research has in fact highlighted a difference in GSS2 scores between neutral and forensic contexts, especially with regards to scores relating to negative feedback. This seems to suggest that in future research it is necessary to also include child witnesses to test the actual effect of IQ, but also of other psychological variables, on interrogative suggestibility.
I hope these comments are helpful and assist in enhancing your paper.
Reply: Sincere thanks because the suggestions were very useful. We also want to point out that the manuscript has been reviewed by an English language expert
Round 2
Reviewer 3 Report
Comments and Suggestions for Authors
I have reviewed the revisions and have no further comments. I believe the authors have addressed my concerns.